# Insights from Structure–Function Studies into Perception of Fatty Acid-Derived Defense Signals

**DOI:** 10.3390/plants14223518

**Published:** 2025-11-18

**Authors:** Johannes W. Stratmann, Harshita Negi, Qian Wang

**Affiliations:** 1Department of Biological Sciences, University of South Carolina, Columbia, SC 29208, USA; hnegi@email.sc.edu; 2Department of Chemistry and Biochemistry, University of South Carolina, Columbia, SC 29208, USA; wang263@mailbox.sc.edu

**Keywords:** bruchins, caeliferins, fatty acid-amino acid conjugates, green leaf volatiles, medium chain fatty acids, sphingoid bases, defense, pattern recognition receptor, volatile organic compounds, Arabidopsis

## Abstract

Studies that correlate the structure of a molecule with its biological function or activity are useful in identifying the structural components that determine how the molecule interacts with binding proteins. This enables the synthesis of structural analogs with desirable properties, such as agrochemicals that improve plant developmental traits or adaptations to environmental stress. This review highlights a group of plant defense-inducing small signaling molecules characterized by a fatty acid-derived molecular skeleton with different functional groups. These include medium chain 3-hydroxy fatty acids (mc-3-OH-FAs) derived from the bacterial cell wall; green leaf volatiles (GLVs), which comprise primary aldehydes, alcohols, and esters derived from plant membranes; insect-derived fatty acid-amino acid conjugates (FACs), caeliferins, and bruchins; and sphingoid bases from oomycete pathogens. These molecules are typically lipophilic, and their mechanism of action is likely determined by both specific structural hallmarks and physicochemical properties. They activate defense responses via signaling pathways and are therefore presumed to interact with extra- or intracellular receptor proteins. However, classical receptors have only been characterized for mc-OH-FAs, sphingoid bases, and FACs. Structure–function studies may reveal structural features of these molecules that are critical for binding to receptors and relevant to the specificity of these interactions. This is particularly significant for GLVs, which have been extensively investigated for their roles in plant stress signaling and interplant communication, yet no specific receptor has been identified to date. This comparative review aims to shed light on perception of GLVs and other small molecules.

## 1. Introduction

The structure of many small molecules that elicit plant defense responses against biotic stressors is based on an aliphatic chain carrying different functional groups. They include free fatty acids, green leaf volatiles (GLVs), fatty acid-amino acid conjugates, (FACs), sphingoid bases, caeliferins, and bruchins (Figure 1). For most of them, it is unknown how they are perceived by plant cells. In receiver cells, they activate defense responses via complex signaling networks, suggesting the existence of a perception mechanism capable of binding these molecules and initiating downstream signaling that culminates in defense gene activation. However, as lipophilic molecules, they may interact with membranes, possibly triggering stress responses via membrane disturbances. In addition, some small molecule elicitors like GLVs can be metabolized and converted to glycosides. For example, in tomatoes, (*Z*)-3-hexenyl acetate can be hydrolyzed to hexenol and then glycosylated to form (*Z*)-3-hexenylvicianoside, which is toxic to lepidopteran larvae [1]. While the metabolization of small molecules may account for their direct toxic effects on herbivores or pathogens, this is unlikely to explain the activation of specific responses such as gene expression.

Structurally conserved elicitors of defense responses can be recognized by plants as molecular patterns [2]. Molecular patterns derived from bacteria and fungi or herbivores are referred to as microbe- or herbivore-associated molecular patterns (MAMPs and HAMPs), respectively, whereas damage-associated molecular patterns (DAMPs) are derived from damaged plant tissue. Many well-characterized molecular patterns interact with pattern recognition receptors (PRRs) [3]. Most PRRs are receptor-like kinases (RLKs) (also known as receptor kinases), which consist of an extracellular ligand recognition domain, a signal transducing transmembrane domain, and an intracellular kinase domain. Binding of molecular patterns to PRRs results in receptor-mediated activation of signaling pathways that induce defense-related gene expression [4]. MAMPs reviewed here include medium chain-3-OH-fatty acids (mc-3-OH-FAs) and sphingoid bases; HAMPs include FACs, bruchins, and caeliferins; and DAMPs include GLVs. Their nature as molecular patterns suggests that they may be perceived by PRRs. This has been confirmed in detailed and elegant studies in *Arabidopsis thaliana* (Arabidopsis hereafter) for mc-3-OH-FAs, which bind to a receptor-like kinase called LIPOOLIGOSACCHARIDE-SPECIFIC REDUCED ELICITATION (LORE) [5,6], and for sphingoid bases, which bind to the lectin receptor-like kinase called RESISTANT TO DFPM-INHIBITION OF ABSCISIC ACID SIGNALING 2 (RDA2) [7]. A high-affinity extracellular binding site for FACs has been characterized in maize [8], and a likely candidate for a maize FAC receptor is the receptor-like kinase FAC SENSITIVITY ASSOCIATED (ZmFACS) [9].

An alternative mode of perception would be by intracellular receptors. Since lipophilic molecules can pass the plasma membrane [10], they could bind to intracellular binding proteins. Examples include five ER-localized histidine protein kinase receptors that bind the stress signal ethylene in a membrane-localized hydrophobic pocket [11], the petunia KAI2 receptor, which perceives the volatile sesquiterpene (-)-germacrene D during flower development and is localized in the nucleus and cytoplasm, as well as other KAI2 receptors that can bind smoke-derived butanolides called karrikins [12,13]. Other small molecules that dock onto intracellular binding proteins include the salicylic acid binding protein SABP2, which binds volatile methyl salicylate and functions as an esterase converting methyl salicylate to salicylic acid, which then binds to its receptor NPR1, leading to activation of gene expression [14]. Another mechanism for intracellular signaling is the binding of small molecules to transcriptional inhibitors in the nucleus. The volatile sesquiterpene β-caryophyllene binds to the transcriptional repressor protein TOPLESS, leading to de-repression of defense gene expression [15]. Unfortunately, this intriguing mechanism of small molecule perception was not further investigated. Finally, ligand-gated ion channels that initiate ion fluxes for signal transduction in animals are often activated by neurotransmitters [16], some of which are also known to function in plant signaling, such as glutamate or γ-aminobutyric acid (GABA) [17,18,19,20]. It is conceivable that similar ion channels could be gated by the group of small molecules discussed in this review.

Some mechanisms of small molecule perception that are well characterized in animals seem to be absent in plants, e.g., plants lack homologs of animal olfactory binding proteins and olfactory receptors that bind odor molecules, including GLVs. Nevertheless, functional analogs of olfactory binding molecules may help to move small molecules around in the cell or the apoplast, such as unspecific lipid transfer proteins [21]. The human sweet receptor is a heterodimer of two seven-transmembrane G protein-coupled receptors. Interestingly, its extracellular binding domain can bind a broad range of structurally diverse small molecules, including various natural sugars and artificial sweeteners, albeit with low affinity (in the millimolar range). This structural flexibility explains why so many different molecules can elicit the sensation of sweetness [22,23]. Such a system may serve as an attractive conceptual model for plant receptors that recognize volatile organic compounds, as it demonstrates how a single receptor can bind structurally diverse yet functionally similar ligands and activate a convergent response. However, no homologs of the human sweet receptor have been identified in plants [24].

A phylogenetic approach can also provide information on the nature of perception mechanisms for small molecules. If molecules are perceived by a narrow taxonomic range of plant species, they are likely to interact with a specific receptor. Similarly, if sensitivity to a molecule is lost in a subclade of an otherwise sensitive clade, this would suggest the presence of a specific receptor that was either not conserved, lost, or functionally altered in the insensitive subclade. Unfortunately, for most of the small molecules covered here, their taxonomic range is poorly understood due to a focus on model species.

In summary, there are numerous ways plants could perceive small signaling molecules. Therefore, it is often very challenging to identify small molecule receptors and demonstrate binding. To build hypotheses and to generate tools for interrogating receptor candidates, a first approach can be structure–function or structure–activity studies. There are many examples where known small signaling molecules were structurally modified to test their effects on biological responses. This can reveal structural features that are critical or dispensable for the activity of a small molecule. In addition, such analyses can lead to useful derivatives that may be inactive or more active, or may function as agonists, antagonists, and competitors of the small molecule of interest. They have advanced the understanding of plant hormone perception and signaling, e.g., for auxin, jasmonic acid, or strigolactones, to name a few [25,26,27]. In analogy to pharmacological drug design, structure–function analyses in plants may also provide a rational basis for predicting and generating novel agrochemicals that boost growth and yield or protect crop plants against environmental stress.

In the following sections, we will review structure–function analyses of eight groups of small molecule elicitors with an aliphatic backbone (Figure 1) to derive conclusions for how structure–function correlations could help to inform modes of perception. By nature of this review, studies focusing on a single small molecule without direct comparison to structurally related molecules are not considered here. Instead, they can be found in several excellent reviews that have covered such studies comprehensively [28,29,30,31,32].

## 2. Medium-Chain Fatty Acids

Medium-chain fatty acids (mcFAs) are derived from lipid A in the inner core of lipopolysaccharides, which are part of the outer membrane of Gram-negative bacteria. Lipid A is made up of acylated glucosamines. The acyl side chains show variability in length (medium to long) and are generally hydroxylated at position 3 or acylated with another fatty acid. In Arabidopsis, it was shown that lipid A is perceived by the SD-lectin receptor kinase LIPOOLIGOSACCHARIDE-SPECIFIC REDUCED ELICITATION (LORE) [6]. A subsequent detailed structure–function analysis demonstrated that free medium-chain 3-hydroxy fatty acids (mc-3-OH-FAs, Figure 1a rather than the entire lipid A are perceived by LORE [5]. A range of structural variants of mcFAs were tested in bioassays measuring their effect on increases in cytosolic Ca^2+^ and reactive oxygen species levels, MAP kinase (MAPK) phosphorylation, and gene expression. It was shown that only 3-OH-FAs with a chain length ranging from 8 to 12 carbons were active, with 10 carbons showing the highest activity, and that the hydroxyl group is critical for activity. Changing its position on the carbon chain or modifying it abolishes activity, and replacing the (*R*)-3-OH group with an (*S*)-3-OH group reduces activity. There was some tolerance for modifying the carboxyl group with short esters and amide-linked groups.

Taken together, this shows that LORE is tolerant to structural modifications of mc-3-OH-C10:0 (C10:0 means ten carbons, zero double bonds), the most active fatty acid, but that the position of the hydroxyl group at the third carbon is critical. There was also no absolute requirement for stereospecificity of the 3-OH-group. In addition, recognition of mc-3-OH-FAs by this type of lectin domain receptor-like kinase specifically evolved in the Brassicaceae family [6], which is a strong sign for the evolution of a specific perception system. Furthermore, mc-3-OH-FA binding by the LORE receptor shows that a relatively lipophilic signaling molecule (XLogP for 3-OH-C10:0 = 2.5), which should be able to pass the plasma membrane, can bind to and signal via an extracellular PRR.

## 3. Green Leaf Volatiles and Derivatives

Non-standard abbreviations used here: hexenal—HAL; hexenol—HOL; hexenyl acetate—HAC.

GLVs are synthesized from linolenic and linoleic acid via a 13-lipoxygenase and a 13-hydroperoxide lyase. The first product, (*Z*)-3-HAL, can be isomerized to (*E*)-2-HAL via an isomerase or spontaneously. The aldehydes can then be reduced by an alcohol dehydrogenase to (*Z*)-3- or (*E*)-2-HOL, which can be converted to (*Z*)-3- or (*E*)-2-HAC by an alcohol acyltransferase (Figure 1b–d). The corresponding three unsaturated GLVs can be synthesized from linoleic acid [33,34]. Together this results in nine core GLVs. In maize, a similar biosynthetic pathway via a specialized lipoxygenase (ZmLox6) generates five-carbon GLVs [35]. GLVs are released from damaged tissue and can induce defense responses in the damaged plant [36,37] and in neighboring receiver plants [28,31]. Since they are structurally defined molecules that are released in response to damage and induce plant defenses, they can be characterized as DAMPs [38,39]. They are best known for their function in responses to herbivory, but they have also been shown to be emitted in response to bacteria, fungi, and abiotic stress [34,40]. It is also important to note that some aldehydes, such as (*E*)-2-HAL, are more reactive than alcohols and esters due to an electrophilic α,β-unsaturated carbonyl bond. (*E*)-2-HAL can react with amino groups, e.g., from amino acids, which correlates with a toxic effect they have on some lepidopteran larvae [41]. This adds a receptor-independent aspect to their perception.

In this section, we will first compare the three different groups of GLVs with regard to their biological effect to determine whether GLV aldehydes, alcohols, and esters elicit different effects on receiver plants. We will then compare structural variants within the three groups to correlate structures with functions. Our analysis will primarily focus on comparative studies that included at least two different GLVs.

### 3.1. Structure–Function Correlations Among the Three GLV Groups

Aldehyde, alcohol, and ester GLVs have been studied extensively, but no receptor or mechanism that activates signal transduction and gene expression has been identified so far. Therefore, structure–function studies may help to define structural components required for binding to an elusive receptor protein. GLV studies were performed in different plant species, addressing different research questions, and using different experimental conditions and approaches. Unfortunately, there are only a few reports directly comparing two or all three groups of GLVs to each other. However, there are several reports on GLV-induced gene expression in *Arabidopsis thaliana* plants. We focused first on these studies to assess whether the three GLV groups may have the same or different effects on receiver plants (Table 1). After all, they are all rapidly (within seconds or minutes) released from damaged tissue [42].

In three studies by Bate and Rothstein [43], Yamauchi et al. [44,45], and Kishimoto et al. [46,47], (*E*)-2-HAL, (*Z*)-3-HAC, hexanol, and (*Z*)-3-HOL, or a subset thereof, induced a similar set of genes related to JA, SA, and biotic stress with some specificity for (*E*)-2-HAL, which also induced expression of genes related to abiotic stress (Table 1). In contrast, Mirabella et al. (2015) [48], found a highly specific gene expression profile for (*E*)-2-HAL, with induced genes enriched in categories related to defense responses to pathogens, especially genes also upregulated by SA (Table 1). Other GLVs, including (*Z*)-3-HAL, (*E*)-2-HOL, (*Z*)-3-HOL, (*E*)-2-HAC, and (*Z*)-3-HAC, did not induce expression of a selection of 12 (*E*)-2-HAL-induced genes.

A high specificity of responses to (*E*)-2-HAL was also shown in three studies that did not involve gene expression analysis. Mirabella et al. (2008) [49] and Scala et al. [50] showed that two mutant genes they identified in a forward genetics screen, *HER1* and *HER2*, exclusively reduce responses to (*E*)-2-HAL [49,50]. 1-hexanol, (*Z*)-3- and (*E*)-2-HOL, and 1-hexanal were not affected by mutations in *HER1* [49] and *HER2* [50]. Furthermore, Aratani et al. showed that (*Z*)-3- and (*E*)-2-HAL, but not n-hexanal, (*Z*)-3-HOL, or (*Z*)-3-HAC, induce Ca^2+^ signaling [51]. (*Z*)-3- and (*E*)-2-HAL (other GLVs not tested) also induced expression of four marker genes, two for heat and oxidative stress, and two for JA-related responses.

For Arabidopsis, the combined comprehensive and diligent works of the Schuurink group [48,49,50] demonstrate that (*E*)-2-HAL activates a specific signaling and gene expression profile that is not shared with the other GLVs. However, when taking all Arabidopsis studies together (Table 1), the results do not allow for a clear conclusion on functional specificity of the three groups of GLVs. When generalizing the functional categories, all three groups are correlated with expression of biotic and abiotic stress-related genes that are mostly, but not always, related to JA signaling. (*E*)-2-HAL seems to stand out for specifically inducing gene expression related to SA-mediated defense responses (Table 1). This conclusion can also be derived from a comprehensive review by Brosset and Blande, who reviewed all known direct and primed responses to volatile organic compounds, including GLVs, until 2021 [31].

Including transcriptomics and proteomics studies from other plant species still does not lead to a clear conclusion on the specificity of the different GLVs (Table 1). In a combined transcriptomics and proteomics study in tomato, (*E*)-2-HAL induced genes and proteins related to pathogenesis and rendered plants resistant to the fungal pathogen *B. cinerea* [52]. This is in line with the Arabidopsis study by Mirabella et al. (2015) [48]. No other GLVs were tested. Other studies in monocot and dicot species showed gene expression in response to (*Z*)-3-HAC, (*Z*)-3-HOL, hexanol, and (*Z*)-3-HAL related to JA-mediated responses to herbivores [53,54,55,56] (Table 1).

Could signaling studies provide deeper insights into GLV specificity? In the grass *Lolium temulentum*, GLVs from cut grass induced MAPK activity similar to leaf wounding [56,57]. No specific GLVs were tested. A recent phosphoproteomics study in *Solanum peruvianum* suspension-cultured cells identified phosphoproteins that are (de-)phosphorylated in response to (*Z*)-3-HOL and (*Z*)-3-HAC and the DAMP (phytocytokine) systemin within five minutes [58]. There was a significant overlap between proteins phosphorylated by all three treatments, indicating a role for these GLVs in wounding- and herbivory-related signaling and response. An almost complete generic DAMP signaling pathway was identified for GLVs, including phosphoproteins like RLKs, a receptor-like cytoplasmic kinase, MAPKs, calcium signaling proteins, and transcription factors. Both GLVs also induced root growth inhibition in tomato seedlings. But there were also differences in the (*Z*)-3-HOL- and (*Z*)-3-HAC-induced phosphorylation profiles, with proteins phosphorylated in response to HOL being dephosphorylated in response to (*Z*)-3-HAC, and seedlings were more sensitive to (*Z*)-3-HAC than to (*Z*)-3-HOL.

**Table 1 plants-14-03518-t001:** Gene and protein expressions induced by C6-GLV aldehydes, alcohols, and esters.

GLVs Tested	Gene Expression Method	Species	Expressed Genes (Functional Categories)	Reference
E2-HAL, hexanol, Z3-HOL, Z3-HAC	10 marker genes	*A. thaliana*	JA biosynthesis and response, phenyl- propanoid synthesis (no defense-pathogens)	Bate and Rothstein, 1998 [43]
E2-HAL Z3-HAL Z3-HOL, Z3-HAC	microarray	*A. thaliana*	abiotic stress, defense (pathogens, SA) defense (pathogens, SA) defense (pathogens, SA), abiotic stress	Yamauchi, 2015 and 2018 [44,45]
E2-HAL, Z3-HAL, Z3-HOL	6 marker genes	*A. thaliana*	JA synthesis and response	Kishimoto, 2005 [46]
E2-HAL, Z3-HAL	5 marker genes	*A. thaliana*	JA synthesis and response (no defense-pathogens)	Kishimoto, 2006 [47]
E2-HAL	microarray & 12 marker genes	*A. thaliana*	defense (pathogens, SA)	Mirabella, 2015 [48]
Z3-HAL, E2-HOL, Z3-HOL, E2-HAC, Z3-HAC	12 marker genes	*A. thaliana*	no response	Mirabella, 2015 [48]
E2-HAL	proteomics, RNAseq	*S. lycopersicum*	defense (pathogens)	Zhang, 2023 [52]
Z3-HAC Z3-HOL, hexanol	3 defense marker genes	*S. lycopersicum*	JA-related	Perez-Hedo, 2021 [53]
Z3-HOL	microarray	*Z. mays*	defense (herbivores), JA synthesis and response	Engelberth 2013 [54]
Z3-HAL Z3-HOL	3 marker genes	*Z. mays*	JA-related JA related	Tanaka 2023 [55]
GLVs from cut grass	RNAseq	*L. temulentum*	defense (herbivores), JA synthesis and response	Dombrowski, 2019 [56]
Z3-HOL, Z3-HAC	phospho- proteomics	*S. peruvianum*	defense (herbivores), DAMP signaling	Tanarsuwongkul, 2024 [58]

Three studies investigated GLV-induced rapid increases in cytosolic Ca^2+^ ions, a common signaling response. But in these studies, different specificities were found, ranging from selective activity of (*Z*)-3- and (*E*)-2-HAL [51] to a range of active GLVs, including (*E*)-2- and (*Z*)-3-HAL, (*E*)-2-HOL [59], and (*Z*)-3-HAC [60]. Different results may be due to different experimental approaches aimed at measuring induced calcium ion fluxes.

The studies reviewed here do not conclusively link the three groups of GLVs to specific responses, e.g., to herbivores, pathogens, or abiotic stress. However, a general conclusion can be drawn from the gene expression and proteomics studies, with the caveat that it is not supported by all studies mentioned here. Each GLV group appears to induce a unique response, however with overlaps between the three groups. Overlaps between alcohols and esters seem to be more pronounced than overlaps between those groups and aldehydes.

Lastly, a few studies looked at combined effects of GLVs. In *Chrysanthemum cinerariaefolium*, the GLVs (*Z*)-3- and (*E*)-2-HAL, (*Z*)-3-HOL, and (*Z*)-3-HAC only induced defense gene expression in combination with (*E*-)-β-farnesene, a terpenoid. None of the five volatile organic compounds (VOCs) was active on its own, showing that in some cases VOCs function synergistically but are inactive when applied alone [61]. Perhaps because single GLVs were active in most other studies, combinatorial effects were not examined and should be investigated in future studies. As shown by Hu et al., even if a single GLV shows an effect, the combined effect of several VOCs may still be synergistic (i.e., above additive) [62]. This adds another level of complexity to the interpretation of VOC perception [63].

### 3.2. Fatty Alcohols

Primary alcohols with aliphatic carbon chains are called fatty alcohols (FAlcs) and include the core GLVs (*Z*)-3- and (*E*)-2-HOL, and hexanol. The IUPAC nomenclature for the (*Z*)-3 isomer of hexenol is (*Z*)-hex-3-en-1-ol. For convenience and in agreement with a large body of the GLV literature, we call this FAlc (*Z*)-3-HOL or Z3-C6:1 in Table 2. Other FAlcs are named accordingly. Saturated FAlcs like hexan-1-ol are named hexanol or C6:0.

We found five studies with six different bioassays that include structure–function correlations for at least two different FAlcs (Table 2). In all studies, (*Z*)-3-HOL was active, except for one small gene expression experiment in Arabidopsis where (*Z*)-3-HOL activity was very low or absent [43]. We do not include reports here that tested only (*Z*)-3-HOL. There are many such reports on (*Z*)-3HOL activity in a range of species, and some of them were mentioned in the prior section.

The saturated FAlc hexanol was tested in five studies. In two of them in maize plants, hexanol was inactive [55,64]. In another monocot, the grass *L. temulentum*, hexanol was active [65], and this was also true for the dicots *A. thaliana* and *S. peruvianum* [43,66]. This indicates a taxon-specific effect of hexanol. The isomers (*E*)-2-HOL and (*E*)-3-HOL were active in monocot species. (*E*)-2-HOL was also active in dicot species. But there is no clear sign that these three FAlcs are generally more or less active than (*Z*)-3-HOL. Since the three mono-unsaturated isomers of HOL were all active in all plant species tested, but hexanol was inactive in maize, there may be a taxon-specific requirement for a double bond in either the C2 or C3 position regardless of *Z*- or *E*-configuration. However, since maize and *L. temulentum*, which both belong to the family Poaceae, showed opposite sensitivity to hexanol, insensitivity may have evolved at the subfamily, genus, or species level. The different bioassays in these studies could also account for the difference.

There is only one study, but with three different bioassays, testing the effects of (*Z*)-3-butenol (Z3-C4:1). It was inactive in two signaling assays in suspension-cultured cells of the wild tomato species *S. peruvianum*, but more active than (*Z*)-3-HOL in a root growth inhibition assay in tomato seedlings [66]. Another shorter chain, FAlc, (*Z*)-2-pentenol (Z2-5:1), was inactive as an inducer of JA-related genes in maize [55]. This indicates that FAlcs with less than six carbons may be generally less active than FAlcs with six or more carbons.

For (*Z*)-3-FAlcs with six to nine carbons, there is a pronounced difference between monocots (maize) and dicots (*A. thaliana* and *Solanum* species). Longer chain FAlcs did not induce expression of three JA-related genes in maize, unlike C6-FAlcs [55], but they induced signaling and root growth inhibition in *S. peruvianum* cells and tomato seedlings [66]. There was a clear chain-length dependence for signaling responses in *S. peruvianum*, showing that an increase in chain length from 4 to 9 carbons correlated with an increase in activity. However, this was less clear for effects of FAlcs on root growth inhibition, where (*Z*)-3-butenol was more active than (*Z*)-3-HOL and (*Z*)-3-octenol slightly more active than (*Z*)-3-nonenol.

In summary, the results for FAlcs with chain lengths ranging from 4 to 9 carbons show a few trends. First, a double bond, regardless of (*Z*)- or (*E*)-configuration, and a chain length of exactly six carbons are critical for bioactivity in maize, but not for the other plants tested. Second, only a subclade of the Poaceae that includes maize is insensitive to hexanol. Third, in the tomato clade, bioactivity of FAlcs correlates with chain length, while saturation state and isomerization seem to play a minor role.

### 3.3. Fatty Aldehydes

Fatty aldehydes (FAlds) are aldehydes with aliphatic chains and include the core GLVs (*Z*)-3- and (*E*)-2-HAL, and hexanal. The IUPAC nomenclature for the (*Z*)-3 isomer of hexenal is (*Z*)-hex-3-enal. For convenience and in agreement with a large body of the GLV literature, we call this FAld (*Z*)-3-HAL or Z3-C6:1 in Table 3. Other FAlds are named accordingly. Saturated FAlds like hexanal are shown as C6:0 in Table 3. We found five studies that include structure–function correlations for FAlds (Table 3). In all studies, (*E*)-2-HAL was active.

Mirabella et al. (2008) [49], Engelberth (2011) [64], and Yamauchi et al. (2015) [44] found that FAld activity in Arabidopsis and maize is chain length-dependent, with an active range of 5–7, 6–8, and 4–9 carbons, respectively. An α, β-unsaturated carbonyl is a distinguishing chemical property of (*E*)-2-FAlds. Various studies tested whether (*Z*)-3-HAL, which lacks this reactive group, is active. The results varied by species and bioassays. (*Z*)-3-HAL was inactive in inducing gene expression in Arabidopsis [44] but active in most other assays, such as JA accumulation in maize [64], induction of rapid increases in cytosolic Ca^2+^-concentrations in Arabidopsis and tomato plants [51,60], and MAPK activation in *L. temulentum* [65]. In contrast to all studies mentioned so far, (*E*)-2-HAL was barely active in inducing extrafloral nectar secretion in lima beans [67]. The double bond for six-carbon FAlds was found to be essential in two Arabidopsis studies [49,51], but not essential for induction of MAPK activity in *L. temulentum* [65].

As for most GLVs, there are additional studies focused on one aldehyde. They were not covered here. For example, nonanal has been studied in more detail. It functions in SA-related resistance against pathogens in bean and barley plants but was not compared to other aldehydes [68,69,70].

### 3.4. Fatty Esters

Fatty esters (FEs) are condensation products of fatty alcohols and acyl-CoA synthesized by alcohol acyltransferases. Modifications of both moieties include the number of carbons, double bonds, position and configuration of double bonds, and branching (Figure 1d). FEs include the core GLVs (*Z*)-3-HAC, (*E*)-2-HAC, and the saturated hexyl acetate. The IUPAC nomenclature for the (*Z*)-3 isomer of HAC is [(*Z*)-hex-3-enyl] acetate. For convenience and in agreement with the GLV literature, we call this FE (*Z*)-3-HAC. All FEs discussed here have a six-carbon alcohol-derived moiety. In Table 4, the isomeric configuration of this moiety and the number of carbons and name of the acyl-CoA-derived moiety are given, as in E2-C2 (acetate) for (*E*)-2-HAC. Other FEs are named accordingly. Different authors used different common names for FEs. Where necessary, we changed the FEs to a common nomenclature, e.g., isovalerate to methylbutyrate.

We found four studies that include structure–function correlations for FEs (Table 4). All compared FEs with a six-carbon alcohol-derived moiety to (*Z*)-3-HAC, which was active in all studies. Therefore, we used (*Z*)-3-HAC as the reference FE in our analysis (Table 4). There were no studies comparing FEs with different lengths of the alcohol-derived moiety. Pentenyl- and pentanyl-derived FEs are known to be synthesized and emitted together with six-carbon GLVs. They play a role in defenses against aphids and fungal pathogens, but no roles for pentenyl esters in defense were determined [71].

**Table 4 plants-14-03518-t004:** Structure–activity relationships for fatty esters.

C6:1 Isomer and Ester Group ^2^	Bioassay	Species	Activity Compared to Z3-C2 (cetate) ^1^	Reference
E2-C2 (acetate)	JA accumulation extrafloral nectar secretion	*Z. mays* *P. lunatus*	> = ^3^ (1) ^5^	Engelberth, 2011 [64] Heil, 2008 [67]
E3-C2 (acetate)	extrafloral nectar secretion	*P. lunatus*	=(1)	Heil, 2008 [67]
5-C2 (acetate)	JA accumulation extrafloral nectar secretion	*Z. mays* *P. lunatus*	inactive <(1,2)	Engelberth, 2011 [64] Heil, 2008 [67]
Z3-C3 (propionate)	defense gene expression pathogen defense ^5^	*S. lycopersicum* *S. lycopersicum*	>(1) >(2)	Pérez-Hedo, 2021 [53] López-Gresa, 2018 [72]
Z3-C4 (butyrate)	extrafloral nectar secretion defense gene expression pathogen defense	*P. lunatus* *S. lycopersicum* *S. lycopersicum*	<(1,2) >(1,2) >(1)	Heil, 2008 [67] Pérez-Hedo, 2021 [53] López-Gresa, 2018 [72]
Z3-C4 (iso-butyrate)	pathogen defense	*S. lycopersicum*	>(3)	López-Gresa, 2018 [72]
C4 (butyrate) ^4^	defense gene expression	*S. lycopersicum*	>(1,2)	Pérez-Hedo, 2021 [53]
Z3-C5 (methyl- butyrate)	extrafloral nectar secretion	*P. lunatus*	<(1,2)	Heil, 2008 [67]

^1^ (*Z*)-3-HAC [Z3-C2 (acetate)] was active in all bioassays. ^2^ All fatty esters tested represent ester groups conjugated to a C6 carbon chain. Isomers referring to the C6 carbon chain and ester groups are indicated by number of carbons and name. ^3^ See Table 2. ^4^ C4 butyrate = hexyl butyrate (saturated). ^5^ Score represents relative activity compared to other FEs in the same experiment, with 1 showing highest activity. (1,2): slightly less active than 1 or slightly more active than 2.

Two studies compared isomers of the alcohol-derived moiety [64,67]. The position of the double bond and its configuration did not have a strong effect on the activity of (*E*)-2-, (*E*)3-, and *5*-HAC, which were similarly active as (*Z*)-3-HAC. However, in one of the two studies, the position of the double bond between the 5th and 6th carbon rendered it inactive [64]. The acyl moiety contained 2 to 5 carbons in the studies we compared (Table 4; Figure 1d). When compared to (*Z*)-3-HAC, all FEs were active in three studies [53,67,72]. FEs with acyl groups containing more than 2 carbons induced slightly lower extrafloral nectar secretion in bean plants [67], and they increased expression of three defense genes [53] and defenses against *P. syringae* [72] in tomato. In the latter study, there was a clear preference for (*Z*)-3-hexenyl butyrate over (*Z*)-3-hexenyl isobutyrate and (*Z*)-3-hexenyl-propionate (Figure 1d), while (*Z*)-3-HAC was the least active FE. This shows that the size of the group generally alters the activity of an FE compared to Z3-HAC, but this can be a reduction or an increase in activity, depending on the study system. Also, the differences between the FEs with acyl moieties containing more than 2 carbons are not striking.

We failed to find studies that determined the effect of different lengths of the alcohol moiety on the activity of FEs, which will be crucial to understand the Structure–function relationship of FEs.

## 4. Other Aliphatic Elicitors of Defense Responses

### 4.1. Fatty Acid–Amino Acid Conjugates

Fatty acid–amino acid conjugates (FACs) are classical HAMPs, which are present in oral secretions of many lepidopteran larvae belonging to various moth clades [73]. They consist of a fatty acid, mostly linolenic (18:3) or linoleic acid (18:2), which is conjugated to either glutamic acid or glutamine. The hydroxylated FAC *N*-(17-hydroxylinolenoyl)-L-glutamine was the first FAC discovered and named volicitin (Figure 1e). It was isolated from oral secretions of the beet armyworm (*Spodoptera exigua*) and shown to induce indirect defense responses in maize plants [74,75].

In the following, we abbreviate the names of FACs based on the number of carbons, double bonds, presence and position of the hydroxyl group of the fatty acid moiety, and the three-letter code for amino acids. For example, N-linolenoyl-L-glutamine is 18:3-Gln and, since one study also considered stereoisomers of Gln, volicitin is shown in Table 5 as 18:3-17-OH-L-Gln, or 18:3-17-OH-D-Gln.

Truitt et al. identified a high affinity extracellular binding site for 18:3-Gln and volicitin in maize membranes with receptor properties [8]. Later, Poretsky et al. found a receptor like kinase (FACS) that confers sensitivity to 18:3-Gln in maize [9]. This could potentially be the receptor representing the binding site described by Truitt et al. [8]. Competitive binding to this FAC binding site was tested with different FACs and correlated with FAC-induced release of VOCs. In contrast to L-volicitin, the stereoisomer D-volicitin was inactive in both the binding and the volatile release assays. In both assays, L-volicitin was more active than 18:3-L-Gln, and the two components of the FAcs, 17-OH-18:3 and Gln, were inactive. This showed that the FAC receptor is stereospecific for the amino acid moiety, that the two unconjugated components of an FAC are inactive, and that the binding site in maize membranes prefers the hydroxylated volicitin over 18:3-Gln.

Ling et al. surveyed FACs in lepidopteran larvae from various species and identified nine different FACs [76]. In a structure–function analysis, they tested four of those for induction of homoterpene VOCs in maize seedlings. These four FACs were added to oral secretions from *S. litura* caterpillars to test an additional effect over the oral secretions due to the added FAC. 18:3-Glu, 18:2-Glu, 18:1-Gln, 16:1-Gln were all active in increasing VOC emissions above emissions induced by oral secretions alone, and the differences between these FACs were minor.

This was further explored by Schmelz et al., who tested the response to volicitin and 18:3-Gln in various plant species [77]. Besides maize, eggplant (*S. melongena*) and soybean (*Glycine max*) responded to volicitin and 18:3-Gln with accumulation of ethylene and jasmonic acid, but not salicylic acid, while tomato, cowpea (*V. unguiculata*), and Arabidopsis did not respond to these FACs. In this paper, a structure–function analysis was performed for soybean. Only volicitin induced ethylene and JA synthesis, while 18:2-Glu, 18:3-Glu, 18:2-Gln, and 18:3-Gln were inactive. In contrast, in maize and eggplant, 18:3-Gln is either slightly more active or as active as volicitin. This shows that the strict preference for volicitin over other FACs in soybean is a species-specific effect, and that sensitivity to FACs does not follow phylogenetic relationships. This was further emphasized by the finding that soybean was sensitive, but cowpea was insensitive to FACs, although both are members of the Fabaceae.

Yoshinaga et al. identified a new FAC from *Manduca sexta* caterpillars, 18-hydroxylineoyl-glutamate (18:3-18-OH-L-Gln), which differs from volicitin only by the position of the hydroxyl group (C-18 instead of C-17) [78]. In maize, the 18-hydroxylated volicitin analog was less active than volicitin, but equally active as 18:3-Gln, while Glu-conjugates of 18:3 were less active. In tobacco and eggplant, hydroxylation of 18:3-Gln at either C17 or C18 rendered these FACs more active than the non-hydroxylated 18:3-Gln.

An extensive phylogenetic study by Grissett et al. tested the correlation between sensitivity to FACs and phylogeny in the Solanaceae family [79]. It included the FACs 18:3-17-OH-Gln (volicitin), 18:3-Gln, and 18:3-Glu in the genera *Solanum*, *Nicotiana*, *Capsicum*, and *Petunia*. Sensitivity was determined using MAPK assays, as MAPK phosphorylation and activation are induced in response to perception of stress signals. This confirmed the results from Schmelz et al. [77], showing that tomato (*S. lycopersicum*) is insensitive, while *S. melongena* (eggplant) is sensitive to FACs. In addition, *C. annuum* (bell pepper), and *Petunia hybrida* and two members of the *Nicotiana* genus, *N. benthamiana* and *N. tabacum*, were able to sense FACs, while three other *Nicotiana* species, and seven members of the tomato clade and *S. tuberosum* (white potato) were not reliably responding to FACs [79]. The strongest response was shown by *S*. *melongena* and *N. benthamiana*, followed by *P. hybrida*, *C. annuum*, and *N. tabacum*. The response to volicitin was tested in a subset of the species mentioned. *S. melongena*, *N. benthamiana*, and *N. tabacum* showed a robust response to volicitin, while *C. annuum* and *P. hybrida* showed no reliable response. In sensitive species, there were slight but no clear differences in the intensity of the MAPK response between 18:3-Gln, 18:3-Glu, and volicitin over a 120 min time course. This analysis shows that perception of FACs is taxon-specific, indicating that the capacity to perceive FACs can be lost and gained, presumably depending on selection pressure of the plant species or recognition of alternative herbivore-related signals. Most species that recognized 18:3-Gln also recognized 18:3-Glu and volicitin, except for *C. annuum* and *P. hybrida*, which were insensitive to volicitin [79].

Other studies identified additional FACs from various lepidopteran species, but did not include structure–function analyses on bioactivity [80,81]. It is also interesting that *N. attenuata* oxidizes 18:3-Glu, resulting in a product with elicitor activity [82].

In summary, a high affinity FAC receptor, most likely a receptor-like kinase such as FACS [9], is located in the plasma membrane of maize cells [8]. It is stereospecific for the amino acid moiety. There seems to be a variety of preferences for FAC modifications in different plant species regarding the amino acid (Gln or Glu) and the length of the fatty acid-derived moiety (C16, C18). The activities of FACs with shorter or longer fatty acid moieties are not known. The amino acid or the fatty acid alone is inactive.

### 4.2. Sphingoid Bases

*Phytophthora infestans*-derived sphingoid bases contain a methylated 18-carbon unsaturated aliphatic chain that carries two alcohol groups (on C1, C3) and one amino group (on C2) (Figure 1f). They are part of sphingolipids, which are components of eukaryotic membranes, and some of them function as MAMPs. In an elegant recent study, Kato et al. showed that sphingoid bases are released from *P. infestans* ceramide (a sphingolipid) by an Arabidopsis ceramidase [7]. The receptor for the resulting sphingoid bases was identified as a lectin receptor-like kinase (RDA2) [7]. A number of structural derivatives of the main sphingoid base (4E,8E,10E)-9-methyl-4,8,10-sphingatrienine (**1**) (Figure 1f) were tested in immune response assays. The sphingoid base (**1**) and a derivative that lacked only the double bond at position 10 (**2**) were similarly active. A derivative that also lacked the methyl group (**3**) was much less active, followed by sphingosine (**4**) (Figure 1f), which lacks both the methyl group and the double bonds at positions 8 and 10. The methyl group at position 9 is characteristic for oomycetes (the clade that contains *Phytophthora*) and fungi, but not for plants. Absence of this methyl group in sphingoid bases (**3**, **4**) resulted in a loss of activity, while the double bond at position 10 (**2**) was less important. Derivatives with a carbon chain of less than 18 carbons were also less active.

Interestingly, the derivatives without the 9-methyl group (**3**, **4**) still bound to the RDA2 receptor in a binding assay with a similar *K*_d_, although their bioactivity in immune assays was much lower than the activity of (**1**) and (**2**).

Together this indicates that the 9-methyl group is not critical for binding of the sphingoid base to the lectin receptor kinase RDA2, but it may be important for activating receptor signaling via conformational effects in the extracellular receptor domain. This is an example of a structure–function study that showed receptor binding and receptor activation can be separated in a small molecule, similar to peptide elicitors [83,84]. It would be interesting to test whether (**3**, **4**) can function as competitors for (**1**, **2**).

### 4.3. Caeliferins

Oral secretions of grasshoppers in the suborder Caelifera (order Orthoptera) contain HAMPs that induce defense responses in maize. Alborn et al. identified disulfooxo-fatty acids as the active elicitors in oral secretions and named them caeliferins [85]. They are either saturated (16:0) or monounsaturated (16:1) sulfated α-hydroxy fatty acids. The ω-carbon carries either a second sulfate group (caeliferin A) or a glycine (caeliferin B) (Figure 1g). Besides these four caeliferins (A 16:0, A 16:1, B 16:0, B 16:1), they also found A17:0 and A18:0. In a maize terpene-induction assay, A16:1 was more active than B16:1, B16:0, A16:0, A17:0, and A18:0, which all had similar activities. So, the changes to the aliphatic backbone had only small effects on caeliferin activity. The sulfooxy, the glycine, or the fatty acid group was not altered or removed, which would be important to gain a better understanding of caeliferin activity and perception. In the aforementioned study by Schmelz et al. [77], it was shown that caeliferin A16:0 induced the synthesis of ethylene and JA in maize and Arabidopsis, but not in cowpea, soybean, tomato, and eggplant.

### 4.4. Bruchins

Doss et al. discovered elicitors in beetles that lay eggs on the seedpods of legumes [86]. The plants respond to the eggs with neoplastic growth that blocks entry of newly hatched beetle larvae into the pod. The active HAMPs in beetle extracts turned out to be esters of unsaturated C22–C24 diols called bruchins (Figure 1h), named after the beetle genus *Bruchus* (family Chrysomelidae). The authors characterized four bruchins (A–D), three of which are bis-3-hydroxypropyl esters, while bruchin A is a monoester [86].

The same group conducted a follow-up study and presented an impressive structure–function analysis with 46 derivatives [87]. In neoplasm induction assays on pea pods, esters of diols with a chain length of 18 to 24 carbons are most active, while a double bond in the chain or its (*E*)/(*Z*)-configuration and position is not relevant for activity. Removing one of the propanoate esters hardly affected activity, but removing an alcohol group strongly reduced the activity of the resulting monoesters of mono-alcohols. It was also shown that incorporating aromatic rings into the carbon backbone left the resulting derivatives active, although less active than the natural bruchins. Modifications on the ester termini also strongly reduced activity. Of the 42 derivatives of the 4 natural bruchins, 25 were at least slightly active, and 14 were as active as the natural bruchins. In summary, critically important for activity are chain length, the two terminal alcohol groups, and the presence and structure of the ester.

## 5. Effect of Physicochemical Properties of Small Molecules on Perception

An important physicochemical feature of small molecule signals is their lipophilicity or hydrophilicity, as it determines if and how a molecule can interact with membranes and thus how and where it can be perceived by a receptor. Widhalm et al. (2015) provided a detailed analysis linking physicochemical properties of volatile molecules to their behavior in various cellular compartments such as intracellular membrane systems, plastids, plasma membrane, cell wall, and cuticle [10]. Their initial model was based on VOC emissions in flowers but was later adapted for leaves [88], greatly increasing our understanding of emission and uptake of volatile compounds, which can be extended to other non-volatile small molecules with few adjustments. A key molecular property is the LogP number (octanol/water partition coefficient), the decadal logarithm of the concentration of a non-ionized molecule in a non-polar solvent (usually octanol) divided by the concentration in water. LogP is independent of the pH. However, lipophilicity is pH dependent, and this is included in LogD values. A decreasing pH results in higher LogD values for acidic molecules, and in lower values for basic molecules [89]. For lipophilic ionizable (acidic) compounds, diffusion through the plasma membrane is possible and may be facilitated by the pH gradient across the plasma membrane with a pH of 5–6 in the cell wall area and a pH of >7 in the cytosol [90]. Solubility should be lower in the acidic cell wall (lower LogD) and higher in the neutral to alkaline cytosol (higher LogD).

The difference between LogP and LogD during phase transitions is negligible for most of the small molecules considered in this review, e.g., for the GLVs. However, for other molecules, we will consider whether pH effects may alter the translocation dynamics of small molecules (Section 6 and Section 7). The software Chemicalize was used for structure generation and prediction of logD and logP (used September 2025, https://chemicalize.com/, developed by ChemAxon (http://www.chemaxon.com); URLs accessed on 12 November 2025)).

Elicitors must strike the right balance between lipophilicity and hydrophilicity to navigate the different cellular compartments on their path to a membrane-bound extracellular or an intracellular receptor. If they enter a leaf via stomata and reach target cells, they first face the aqueous cell wall, followed by the plasma membrane and then the aqueous cytosol, and perhaps intracellular membrane systems. The plasma membrane constitutes the highest resistance to diffusion [10], and lipophilic molecules may be concentrated in the membrane [88], unless they are transported by endocytosis or dedicated, so far unknown, transporters. Most known transporters of fatty acids and lipids export cargo from plastids into the cytosol or from the cytosol into the ER or apoplast, but not from the cell wall into the cytosol [91]. Elicitors that diffuse through membranes must also navigate different pH environments.

## 6. Perception of Small Molecule Elicitors by Membrane-Bound Receptors

There are three examples for receptors of small molecule elicitors with an aliphatic backbone as follows: LORE for mc-3-OH-FAs, RDA2 for sphingoid bases, and FACS for FACs. For all three, perception is LogP-independent.

Active mc-3-OH-FAs (8 to 12 carbons) have LogP values ranging from 1.4 to 3.6 (2.5 for the most active FA with 10 carbons). Longer 3-OH-FAs have higher LogPs and are inactive. Also, the active (*R*) stereoisomer of the 3-OH group has the same LogP as the less active (*S*) isomer. This shows that for mc-3-OH-FAs, binding to the RLK LORE and bioactivity are independent of lipophilicity. Since GLVs with a comparable LogP value, such as (*Z*)-3-HOL, are known to pass through the plasma membrane [1], a 10-carbon 3-OH-FA should also be able to do so. And this would be favored by the cytosolic pH. The mc-3-OH-FAs are medium-strong acids. Therefore, in the extracellular environment of the cell wall, the LogD [pH5] for 3-OH-10:0 is ~2, and in the environment of the cytosol, the LogD [pH7] is ~0, leading to a higher solubility in the cytosol than predicted by the LogP. Yet, mc-3-OH-FAs are perceived by an extracellular receptor, the lectin-receptor kinase LORE. The fatty acid-amino acid conjugate volicitin is an amphiphilic molecule with a LogP of 3.2. Its LogD at pH 5 and 7 are very similar to the 10-carbon 3-OH-FA. Yet, there is strong evidence for an extracellular high-affinity binding site on maize plasma membranes [8]. Also, the loss of sensitivity to FACs in some clades of the Solanaceae is in line with rapid evolution of specific receptors.

Sphingoid bases generally have long carbon chains (C12 to C19 tested), two hydroxyl groups and an amino group. The LogP value for (4E,8E,10E)-9-methyl-4,8,10-sphingatrienine (Figure 1f) is 4.1, which is higher than the LogP numbers for the active 3-OH-FAs (1.3–3.6). Indeed, as a component of ceramide and other sphingolipids, they are integral components of eukaryotic membranes, and they also serve in various signaling processes in plant and mammalian cells [92]. However, as bases, they are more soluble in the lower pH environment of the cell wall (LogD [pH5] ~1), which may facilitate interaction of elicitor-active sphingoid bases with their extracellular receptor RDA2.

These three examples clearly show that lipophilicity and extra- or intracellular perception of a small molecule are not necessarily linked. Another well-known example of a lipophilic signal is brassinolide (LogP 4.8), which is perceived by the LRR-RLK BRASSINOSTEROID INSENSITIVE 1 (BRI1) [93].

RDA2 (At1g11330) and LORE (AT1G61380) are close homologs [5,7] containing the same ectodomains: two lectin domains (B-lectin and SLG) and a plasminogen–apple–nematode domain (PAN), and their ligands both have medium to long carbon chains and polar functional groups. Therefore, it could be hypothesized that lectin-RLKs may be ideal for binding such molecules. However, when we analyzed hydropathy plots of their ectodomains (at SUBA5; https://suba.live; accessed on 12 November 2025), the two lectin RLKs showed no pronounced hydrophobic domain that might interact with the carbon chains of sphingoid bases or FAs. Either the unknown binding site for these ligands requires only a short sequence of hydrophobic amino acids that is not obvious from hydropathy plots, or it binds with the hydrophilic functional group, and the carbon chain may contribute to activation of the receptor after binding. For both ligands, the chain length is critically important for bioactivity, so the carbon chain does play a role in ligand perception. If the LRR-RLK FACS is the receptor for FACs, similar considerations apply. Since leucine-rich repeats (LRRs) exhibit a hydrophilic surface, the amino acid moiety of an FAC may bind to LRRs, and the fatty acid moiety would bind to a non-LRR region, similar to brassinolide, which binds to an island domain within the LRRs of the BRI1 ectodomain [93]. Unfortunately, with so few receptors known for small molecule elicitors with aliphatic side chains, it is difficult to predict how ligand-receptor interactions are shaped by the hydropathy of the receptor ectodomains.

## 7. Unknown Mechanisms of Perception of Small Molecule Elicitors

For the three groups of GLVs, FACs, caeliferins, and bruchins, no receptor has been identified yet. By comparing structure–function relationships for these molecules to the three with a cognate receptor, we may be able to obtain insights on their mode of perception.

### 7.1. Fatty Alcohols (FAlcs)

There was a clear difference between maize and tomato in sensitivity to FAlcs. In maize, only E3- and Z3-HOL were active in a gene expression assay, while FAlcs with more or less carbons than HOLs were inactive [55]. In tomato, there was a correlation between chain length and activity, with a lesser role for saturation state and isomerization [66]. These data suggest several possible perception mechanisms. (A) The narrow range of active FAlcs and the taxon-specificity in maize plants indicate a specific perception system in maize, such as a PRR. (B) This would be different from FAlc perception in other plants. The correlation of chain length and activity in the tomato clade [66] is compatible with various modes of perception. While Fisher et al. did not include FAlcs with more than 9 carbons [66], unpublished data by our group indicate that 250 µM dodecanol (12:0) exhibited a lower and slower pH response in *S. peruvianum* cells than both nonanol (C9:0) and (*Z*)-3-nonenol (C9:1) (Stratmann and Fisher, unpublished results). With butenol and dodecanol being inactive or much less active than C5 to C9 FAlcs, the tomato response to FAlcs resembles the activation of the Arabidopsis RLK LORE by a range of mc-3-OH-FAs in [5]. Also, the most active mc-3-OH-FA (C10:0) is more lipophilic (LogP 2.5) than (*Z*)-3-HOL (LogP 1.3). Therefore, lipophilicity would not prevent (*Z*)-3-HOL from binding to an extracellular receptor. Furthermore, the (*Z*)-3-HOL-induced phosphoproteome pattern in *S. peruvianum* cells is in line with the presence of a membrane-bound PRR-like HOL receptor [58]. (C) The lipophilicity-activity correlation for FAlcs may also indicate a mechanism that includes diffusion or transport through the plasma membrane and interaction with an intracellular broad-specificity receptor. (D) Since FAlcs are not acids or bases, their LogP is pH-independent. Therefore, transition of the hydrophobic FAlc from cell wall to membrane should occur at a low rate. However, the Dudoreva group calculated that the (*Z*)-3-HOL transition from air to cytosol via the cell wall and plasma membrane can happen solely by diffusion and does not require specific transport mechanisms [88]. This study provides important parameters for predicting where VOCs may interact with receptors and leaves open the possibility that (*Z*)-3-HOL may activate intracellular receptors.

### 7.2. Fatty Aldehydes (FAlds)

Studies shown in Table 3, indicate that an FAld perception mechanism has a limited tolerance for carbon chain length, that the double bond in position 2 or 3 is necessary for bioactivity, but that the α, β-unsaturated carbonyl group is most likely not essential. E2-HAL, nonanal, and decanal were barely active or inactive in inducing extrafloral nectar secretion in lima bean [67], which contrasts with most other studies. This may indicate that extrafloral nectar secretion is not part of the response to FAlds in lima bean plants (*P. lunatus*; Fabaceae), which is an indirect defense response against herbivores. In common bean plants (*P. vulgaris*), nonanal upregulates pathogenesis-related genes and induces resistance against pathogens [69]. This shows that nonanal is capable of inducing a fairly specific response. A response-specific effect is hard to reconcile with an unspecific FAld perception mechanism. In addition, if lipophilicity were to be a primary determinant for activity, longer FAlds should be more and not less active, in contrast to what was shown in three of the studies shown in Table 3. Furthermore, the highly specific (*E*)-2-HAL-induced gene expression profile and the identification of two genes, *HER1* and *HER2*, which exclusively affect responses to (*E*)-2-HAL, strongly suggest a specific perception mechanism for FAlds [48,49,50].

### 7.3. Fatty Esters (FEs)

Due to the paucity of structure–function studies for FEs, it is difficult to speculate on an FE perception mechanism. For an FE with a C6-alcohol-derived moiety, replacing the acetate moiety of (*Z*)-3-HAC had relatively mild effects on activity. More studies are required to determine whether the isomeric configuration contributes to activity. The key may be the alcohol-derived moiety, but variations thereof were not included in any of the comparative structure–function studies. The Tanarsuwongkul et al. study showed that (*Z*)-3-HAC rapidly induced a DAMP-like phosphoproteome profile and MAPK activation, which indicates a specific mode of perception via a PRR-like receptor [58].

### 7.4. All GLVs

We can now discuss whether responses to FAlcs, FAlds, and FEs are mediated by one or more receptors. The structure–function studies showed that at least FAlcs and FAlds are likely to be perceived by specific receptors. For FEs, the evidence is weaker due to a lack of studies. Their similar yet distinct effects, their release in response to cellular damage, and their activation of signaling pathways suggest that the three groups of GLVs are perceived by at least three PRR-like receptors. In the comparative studies we analyzed, overlaps between FAlcs and FEs are more pronounced than overlaps between those groups and FAlds. Therefore, FAlc and FE receptors may generate a similar signaling pattern and defenses against herbivores, while FAlds generate a different pattern inducing responses to pathogens and abiotic stress. This would not be unprecedented. The structure of molecular patterns is widely different, and the PRRs that perceive them are highly specific, but receptor-like cytoplasmic kinases can channel signals from different PRRs and generate similar signaling patterns leading to similar output responses [94]. On the other hand, different molecular patterns can induce different defense responses, e.g., systemin and flg22 are both peptides perceived by LRR-RLKs (SYR1 and FLS2), but systemin activates jasmonic acid signaling leading to defenses against herbivores [95,96,97,98], while flg22 activates salicylic acid signaling and defenses against pathogens [99].

Heil et al. (2008) [67,100] suggested an alternative way for how GLVs may induce defense responses that does not depend on specific receptors. They proposed that diffusion of GLVs into the plasma membrane would induce changes in the membrane potential through effects on ion permeability, eventually resulting in gene expression. This would be a relatively unspecific signaling mechanism, which is difficult to reconcile with specific signaling responses such as MAPK activation, Ca^2+^- and H^+^-fluxes, and induction of specific defense responses such as priming, gene expression, and VOC emissions. It is important to note that for universal signaling responses like MAPK activation and Ca^2+^ fluxes, specificity is achieved by generating input-specific signaling patterns generated by a combination of multiple factors, such as differential activities of various homologs (e.g., the MAPKs MPK6, MPK4, and MPK3 in Arabidopsis), cell type, subcellular localization, kinetics (activation, duration/downregulation), amplitude, and interactions between different signaling components [101]. For instance, MAPK activation occurs via a highly coordinated cascade of MAPK and MAPK, with each kinase being part of a multigene family. Furthermore, MAPK cascades are generally activated either directly by membrane-bound or intracellular receptors or via an intermediate receptor-activated receptor-like cytoplasmic kinase (RLCK) [101]. Taken together, unspecific interactions of lipophilic small molecules may generate signals such as ion fluxes and membrane de- or hyperpolarizations [60,67], but it is unclear how that would translate into output-specific signaling patterns. Also, it is often difficult to distinguish disturbance effects from specific signal transduction events. Dose–response correlations are critical here. If bioactivity is limited to high concentrations of an elicitor molecule (µM to mM), this may indicate unspecific effects, but it could also point to the presence of low-affinity receptors.

### 7.5. Caeliferins

The LogD of caeliferins strongly depends on the pH. Caeliferin A16:0 has a LogD [pH5] of −2 and a LogD [pH7] of −3.8, so it is highly soluble in the cell wall and the cytosol. Therefore, passing the plasma membrane barrier for perception by an intracellular receptor would require special transport mechanisms. In addition, since only select species within mono- and dicots are sensitive to caeliferin [77], perception is likely to require a specific extracellular receptor. This is similar to FACs, which are also perceived in a taxon-specific manner and bind to a specific receptor. Not enough studies were conducted to confidently predict the nature of a caeliferin receptor. However, if there is only one caeliferin receptor, it is predicted to tolerate major differences at its ω-carbon, such as sulfate or glycine groups.

### 7.6. Bruchins

Oliver et al. (2002) showed that bruchins from beetle eggs contain structural features that were shown to be important for bioactivity [87]. By and large, their structure–function analysis indicates the presence of a receptor with a relatively broad tolerance for structural modifications of the natural bruchins, exceeding the flexibility of the LORE receptor. Since bruchins are not ionizable (neither acid nor base), their LogP is independent of pH. They represent the most lipophilic molecules shown in Figure 1, which may diffuse through the plasma membrane and interact with an intracellular receptor. On the other hand, FACs, mc-3-OH-FAs, and sphingoid bases show that lipophilicity does not exclude perception by an extracellular receptor. In the case of bruchins, the solubility in the cell wall is extremely low, which may make an interaction with an extracellular receptor difficult. The lipophilicity may enable bruchins to diffuse through the waxy cuticle directly underneath a deposited egg to induce neoplastic growth initiated by the epidermal cell layers at the egg deposition site [102].

## 8. Conclusions and Perspectives

(a) Many reviews on the perception of VOCs discuss VOCs as a coherent group of molecules based on their volatility rather than as structurally diverse molecules. However, perception is most likely determined by structure and physicochemical properties. Regardless of whether a defense molecule travels in the gas phase, the vasculature, or is derived from herbivores, pathogens, or damaged cells, it first arrives in the apoplast/cell wall area. Based on their structure, they will then bind to extracellular receptors or be trafficked or diffuse through the membrane for intracellular perception.

(b) Physicochemical properties like lipophilicity, solubility, and volatility will add constraints on the mode of perception, but it has been demonstrated that small lipophilic molecules like mc-3-OH-FAs, sphingoid bases, and FACs, can be perceived in the aqueous apoplastic space by extracellular receptor domains.

(c) Most small molecules described here are probably perceived by specific receptors because they exhibit specific structure–activity correlations within a limited range of structural variations. However, their subcellular localization cannot be determined by structure–activity studies. Since the molecules reviewed here are all molecular patterns that induce defense responses such as defense gene expression and signal transduction, and three groups of them are recognized by known PRRs, the remaining groups may also signal via PRRs. However, intracellular receptors similar to the ethylene or germacrene receptors, or perception by transcriptional regulators, cannot be excluded (see Introduction).

(d) Lipophilicity-activity correlations may point to unspecific interactions with membranes, which play an important role in signal perception. However, it is not known how unspecific effects on membranes would induce specific signaling patterns and responses.

(e) Taxon-specific sensitivity to small molecules indicates the presence of a specific receptor, as demonstrated for the RLKs FACS for FACs and LORE for mcFAs. RLKs can be organized in chromosomal clusters, probably due to frequent gene duplications. This could explain taxon-specific neofunctionalization of an RLK, or loss of an RLK in a taxon for which it is of little adaptive value [103,104]. Interestingly, for FAlcs, a taxon-specific effect was observed with regard to the specificity of perception. Hexenol sensitivity is shared by all plants tested, but the sensitivity in maize is highly specific to (*E*)-2- and (*Z*)-3-hexenol, whereas additional FAlcs are active in other species (Table 2). This indicates the presence of different hexenol receptors in different groups of plants.

(f) For many small molecules, structure–activity studies are rare. In addition, a comparison of different studies is often challenging because they include different species, bioassays, derivatives, and experimental conditions. Since these studies can be highly informative, they should be carried out in a variety of species, using standardized tests for direct comparison of similar molecules, and including a range of derivatives that allow for testing of various structural parts of a molecule, e.g., for FEs, a study comparing the length of the fatty acid-derived moiety is missing.

(g) A fine-tuned structure–activity study would be provided by omics analyses, e.g., transcriptomics, proteomics, or phosphoproteomics. Comparing related small molecules at this level and under standardized conditions has the potential to unravel major as well as subtle differences in the response to related molecules. Including a well-known molecular pattern in the same study would be helpful as a reference for a comparative analysis [58]. The drawback is that this is very costly for extensive structure–activity studies. However, comparing representative molecules from different small molecule groups plus a known molecular pattern, would be highly informative and less expensive.

(h) To identify receptors for small molecules, many techniques are available. For biochemical identifications, labeling techniques are often difficult because of the small size of the molecules and relatively big tags, such as fluorescent, photoaffinity, or biotin tags. Minitags combined with click chemistry are a promising approach to overcome this problem [105,106]; however, for volatile molecules like GLVs, tagging may reduce volatility and would depend on liquid systems such as liquid-grown seedlings, protoplasts, or suspension-cultured cells. Forward and reverse genetic approaches are very powerful. The hexenal-specific *HER1* and *HER2* genes as well as the receptor genes *LORE*, *RDA2*, and *FACS* were all identified using forward genetics, facilitated by powerful whole-genome based mapping techniques [5,7,9,49,50]. When insensitive genetic variants can be identified, e.g., in ecotypes or introgression lines between genetically compatible species, forward genetics can be accelerated by focusing on the known differences between the variants [97]. Candidates identified through omics studies can be further tested for insensitivity using mutagenesis.

(i) A conceptually new way to think about perception of volatile small molecules was pioneered by the Blande group in research on secondary organic aerosols (SOAs) [107]. When GLVs and other VOCs are oxidized in the atmosphere, e.g., by pollutants such as ozone and nitric oxide, they form SOA particles that can interact with receiver plant surfaces, potentially resulting in locally high concentrations of active VOCs. The oxidized products in SOAs could disturb perception and signaling of undamaged VOCs, leading to unspecific responses. However, Yu et al. show that the response to SOAs can be specific [107]. SOAs derived from herbivore-induced VOCs were shown to prime defenses and induce resistance against pine weevils in Scots pine (*Pinus sylvestris*). Therefore, the emitted VOCs may have been chemically altered in the atmosphere before arriving at receiver plants as SOAs. It remains to be determined which chemicals in SOAs are active as plant defense inducers, and if SOAs can enter the leaf cavity through stomata and then transition to the cell wall for delivery of the modified small molecules, possibly followed by passage through the membrane and into the cytosol. Discovery of herbivore-induced VOCs in SOAs opened a new perspective on interplant communication with transformative implications for VOC perception. Most of the work on SOAs from plant VOCs is focused on terpenoids that oxidize in air and form SOAs [108], but ozonolysis of GLVs can turn (*Z*3)-HAC or (*Z*3)-HOL into 3-acetoxypropanal and 3-hydroxypropanal, respectively, and then into additional oxidation products that all form SOAs [109,110,111]. Hydroxyl and nitrate radicals can also oxidize GLVs. It would be exciting to test major GLV oxidation products from natural and artificially generated SOAs for effects on plant defenses and signal transduction.

## Figures and Tables

**Figure 1 plants-14-03518-f001:**
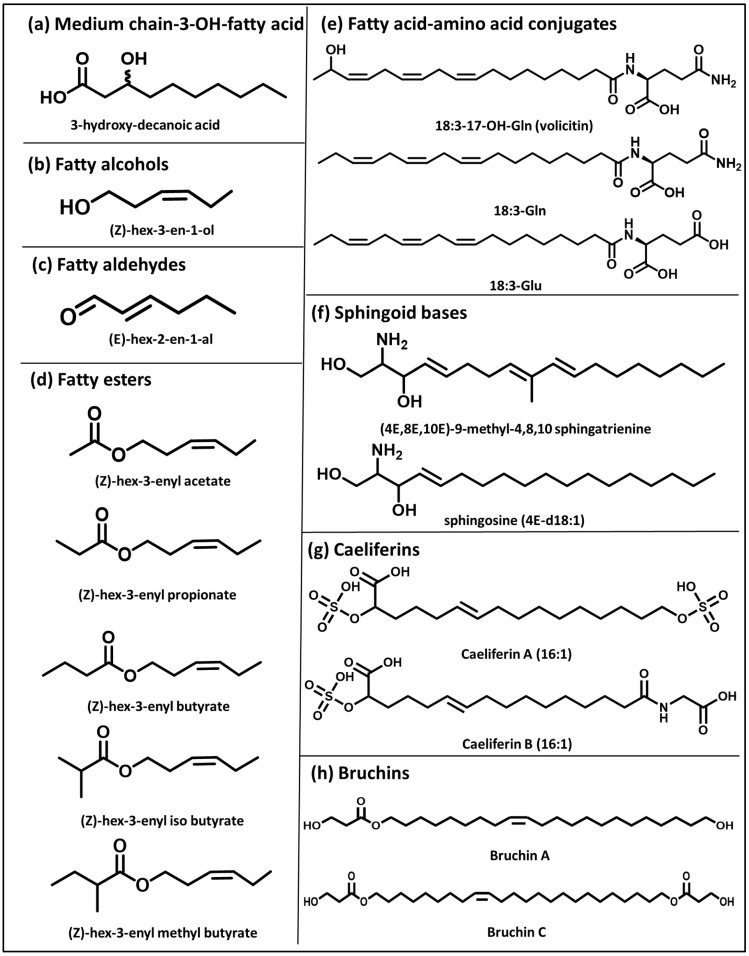
Structures of eight classes of small aliphatic defense molecules. (**a**) medium chain-3-OH-fatty acids; (**b**) fatty alcohols; (**c**) fatty aldehydes; (**d**) fatty esters; (**e**) fatty acid-amino acid conjugates; (**f**) sphingoid bases; (**g**) caeliferins; (**h**) bruchins. GLVs are shown in subfigures (**b**–**d**).

**Table 2 plants-14-03518-t002:** Structure–activity relationships for fatty alcohols.

Falc ^2^	Bioassay	Species	Activity Compared to Z3-C6:1 ^1^	Reference
C6:0	JA accumulation JA-related genes MAPK activation LOX-1 gene expr. pH change	*Z. mays* *Z. mays* *L. temulentum* *A. thaliana* *S. peruvianum*	inactive inactive = ^3^ > >	Engelberth, 2011 [64] Tanaka, 2023 [55] Dombrowski, 2018 [65] Bate, 1998 [43] Fisher, 2025 [66]
E2-C6:1	MAPK activation LOX-1 gene expr. pH change	*L. temulentum* *A. thaliana* *S. peruvianum*	= = >	Dombrowski, 2018 [65] Bate, 1998 [43] Fisher, 2025 [66]
E3-C6:1	JA-related genes MAPK activation	*Z. mays* *L. temulentum*	< =	Tanaka, 2023 [55] Dombrowski, 2018 [65]
Z3-C4:1	pH change MAPK activation root growth inhibition	*S. peruvianum* *S. peruvianum* *S. lycopersicum*	inactive inactive >	Fisher, 2025 [66] Fisher, 2025 [66] Fisher, 2025 [66]
Z2-C5:1	JA-related genes	*Z. mays*	inactive	Tanaka, 2023 [55]
Z3-C7:1	JA-related genes pH change MAPK activation root growth inhibition	*Z. mays * *S. peruvianum* *S. peruvianum* *S. lycopersicum*	inactive > (<C8:1/C9:1) > (<C8:1/C9:1) > (<C8:1/C9:1)	Tanaka, 2023 [55] Fisher, 2025 [66] Fisher, 2025 [66] Fisher, 2025 [66]
Z3-C8:1	JA-related genes pH change MAPK activation root growth inhibition	*Z. mays * *S. peruvianum* *S. peruvianum* *S. lycopersicum*	inactive > (>C7:1; <C9:1) > (>C7:1; <C9:1) > (>C7:1/C9:1)	Tanaka, 2023 [55] Fisher, 2025 [66] Fisher, 2025 [66] Fisher, 2025 [66]
Z3-C9:1	JA-related genes pH change MAPK activation root growth inhibition	*Z. mays * *S. peruvianum* *S. peruvianum* *S. lycopersicum*	inactive > (>C7:1/C8:1) > (>C7:1/C8:1) > (>C7:1; <C8:1)	Tanaka, 2023 [55] Fisher, 2025 [66] Fisher, 2025 [66] Fisher, 2025 [66]

^1^ (*Z*)-3-HOL (Z3-C6:1) was active in all bioassays. ^2^ FAlc (fatty alcohol) species are shown by isomer configuration and number of carbons and double bonds. ^3^ The equal sign (=) indicates that no difference to Z3-C6:1 was reported; >, < signs: more or less active than Z3C6:1.

**Table 3 plants-14-03518-t003:** Structure–activity relationships for fatty aldehydes.

Fald ^2^	Bioassay	Species	Activity Compared to E2-C6:1 ^1^	Reference
C6:0	root growth inhibition Ca^2+^-fluxes MAPK activation	*A. thaliana* *A. thaliana* *L. temulentum*	inactive inactive = ^3^	Mirabella, 2008 [49] Aratani, 2023 [51] Dombrowski, 2018 [65]
Z3-C6:1	JA accumulation *HSFA2* TF expression Ca^2+^-fluxes MAPK activation	*Z. mays* *A. thaliana* *A. thaliana* *L. temulentum*	< (>C8:1) inactive > =	Engelberth, 2011 [64] Yamauchi, 2015 [44] Aratani, 2023 [51] Dombrowski, 2018 [65]
C3:1 (acrolein)	JA accumulation *HSFA2* TF expression	*Z. mays * *A. thaliana*	inactive inactive	Engelberth, 2011 [64] Yamauchi, 2015 [44]
E2-C4:1	JA accumulation *HSFA2* TF expression	*Z. mays * *A. thaliana*	inactive =	Engelberth, 2011 [64] Yamauchi, 2015 [44]
E2-C5:1	*HSFA2* TF expression root growth inhibition	*A. thaliana* *A. thaliana*	= < (>C7:1)	Yamauchi, 2015 [44] Mirabella, 2008 [49]
E2-C7:1	*HSFA2* TF expression root growth inhibition	*A. thaliana* *A. thaliana*	= < (<C5:1)	Yamauchi, 2015 [44] Mirabella, 2008 [49]
E2-C8:1	JA accumulation *HSFA2* TF expression	*Z. mays* *A. thaliana*	< (<Z3-6:1) >	Engelberth, 2011 [64] Yamauchi, 2015 [44]
C9:0	MAPK activation	*L. temulentum*	=	Dombrowski, 2018 [65]
E2-C9:1	JA accumulation *HSFA2* TF expression root growth inhibition	*Z. mays* *A. thaliana* *A. thaliana*	inactive = inactive	Engelberth, 2011 [64] Yamauchi, 2015 [44] Mirabella, 2008 [49]
C10:0	MAPK activation	*L. temulentum*	=	Dombrowski, 2018 [65]
E2-C10:1	*HSFA2* TF expression	*Z. mays*	inactive	Yamauchi, 2015 [44]

^1^ (*E*)-2-HAL (E2-C6:1) was active in all bioassays. ^2^ FAld (fatty aldehyde) species are shown by isomer configuration and number of carbons and double bonds. ^3^ See Table 2.

**Table 5 plants-14-03518-t005:** Structure–activity relationships for fatty acid–amino acid conjugates.

FAC	Bioassay	Species	Activity, Ranked ^1^	Reference
18:3-17-OH-L-Gln ^2^ 18:3-17-OH-D-Gln 18:3-L-Gln Gln (no FA) 18:3-17-OH (no amino acid)	competition for [^3^H]-L-18:3-OH binding	*Z. mays * (membrane preps)	1 no competitor 2 no competitor no competitor	Truitt, 2004 [8]
18:3-17-OH-L-Gln 18:3-17-OH-D-Gln 18:3-L-Gln Gln (no FA) 18:3-17-OH (no amino acid)	VOC release assay	*Z. mays * (seedlings)	1 inactive 2 inactive inactive	Truitt, 2004 [8]
18:3-Glu 18:2-Glu 18:1-Gln 16:1-Gln	VOC release assay (when added to oral secretions)	*Z. mays * (seedlings)	active active active active	Ling, 2021 [76]
18:3-17-OH-L-Gln 18:3-Gln	ethylene and JA synthesis	*Z. mays*, *S. melongena*, *G. max*	active active	Schmelz, 2009 [77]
18:3-17-OH-L-Gln 18:3-Gln	ethylene and JA synthesis	*A. thaliana*, *V. unguiculata*, *S. lycopersicum*	inactive inactive	Schmelz, 2009 [77]
18:3-17-OH-L-Gln 18:3-Gln 18:3-Glu 18:2-Gln 18:2-Glu	ethylene and JA synthesis	*G. max*	active inactive inactive inactive inactive	Schmelz, 2009 [77]
18:3-17-OH-L-Gln 18:3-18-OH-L-Gln 18:3-Gln 18:3-Glu 18:3-18-OH-L-Glu	VOC release assay	*Z. mays*	1 2 2 3 3	Yoshinaga, 2014 [78]
18:3-17-OH-L-Gln 18:3-18-OH-L-Gln 18:3-Gln 18:3-Glu 18:3-18-OH-L-Glu	VOC release assay	*S. melongena, N. tabacum*	1 1 2 2 2	Yoshinaga, 2014 [78]
18:3-17-OH-L-Gln 18:3-Gln 18:3-Glu	MAPK activation	seven species in the tomato clade	insensitive insensitive insensitive	Grissett, 2020 [79]
18:3-17-OH-L-Gln 18:3-Gln 18:3-Glu	MAPK activation	*N. alata*, *N. * *sylvestris*, *N. knightiana*	insensitive insensitive insensitive	Grissett, 2020 [79]
18:3-17-OH-L-Gln 18:3-Gln 18:3-Glu	MAPK activation	*N. tabacum*, *N. benthamiana*	sensitive sensitive sensitive	Grissett, 2020 [79]
18:3-17-OH-L-Gln 18:3-Gln 18:3-Glu	MAPK activation	*C. annuum*	insensitive sensitive sensitive	Grissett, 2020 [79]
18:3-17-OH-L-Gln 18:3-Gln 18:3-Glu	MAPK activation	*S. melongena*	sensitive sensitive sensitive	
18:3-17-OH-L-Gln 18:3-Gln 18:3-Glu	MAPK activation	*P. hybrida*	insensitive sensitive sensitive	Grissett, 2020 [79]

^1^ FACs were ranked for each assay; for studies focused on phylogenetic analyses, rankings are not possible, and it is only stated whether a species is sensitive or insensitive to a specific FAC. ^2^ 18:3-17-OH-L-Gln = L-volicitin = volicitin.

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
