# Peer review of "Insights from Structure–Function Studies into Perception of Fatty Acid-Derived Defense Signals"

_plants, 2025, doi:10.3390/plants14223518_

Round 1

Reviewer 1 Report

Comments and Suggestions for Authors

In this article, Stratmann et al demonstrated that plant perception of fatty acid-derived small molecules is intricately linked to their structural features and physicochemical properties, influencing their interaction with specific receptors or cellular components. While some receptors have been identified (e.g., LORE for medium-chain 3-hydroxy fatty acids, RDA2 for sphingoid bases, and FACS for FACs), many remain elusive, especially for GLVs. The review emphasizes that perception often involves both extracellular receptor-like kinases as well as potential intracellular receptors, and that lipophilicity and molecular configuration critically affect how these molecules are perceived and trigger downstream signaling. The work is interesting; however, I have some suggestions:

  1. What are the specific receptors and molecular mechanisms through which different classes of GLVs (aldehydes, alcohols, esters) are perceived in various plant species?
  2. Provide clearer insight or hypotheses on potential intracellular receptors or transport mechanisms for lipophilic fatty acid derivatives.
  3. How do structural variations within fatty acid-derived molecules influence their binding affinity and activation of distinct receptor types?
  4. What is the role of lipophilicity and molecular charge state in determining the cellular localization and perception pathways of these small molecules?
  5. Include more detailed discussions on the physiological consequences of receptor-mediated perception beyond initial signaling, such as metabolic and growth trade-offs.
  6. Suggest future directions employing advanced omics and mutagenesis to identify unknown receptors and signaling components.
  7. How do plants integrate overlapping yet distinct signaling pathways induced by various fatty acid-derived molecules to coordinate specific defense responses?
  8. The role and identity of intracellular receptors or transport mechanisms for lipophilic fatty acid derivatives are only briefly touched upon and merit deeper exploration.

Reviewer 2 Report

Comments and Suggestions for Authors

The manuscript submitted by Stratmann and colleagues reviews what is known about the perception of plant-defense inducing small molecules, the reactions they induce, the known receptors and whether they can pass the plasma membrane based on their structural and physicochemical properties. The authors cover eight classes of small defence molecules and structure the manuscript accordingly, considering their function in plants, how they may be perceived, and the effect of their lipophilicity and pH on their interaction with membranes. 

Overall, the authors provide an in-depth description of the current state of research and conduct additional analyses of structure elucidation and predictions of the octanol/water partition coefficient and the influence of pH on lipophilicity. While the work is interesting, the manuscript could be substantially improved to make it more concise. Currently, it is somewhat repetitive, with some sections covered by other reviews (e.g. Arimura and Urimura, 2025; Bergmann et al., 2025; Loreto and D'Auria, 2022). 

My comments in detail:

  • Based on the title, I expected a different type of manuscript. The title suggests a focus on GLVs, but much of the manuscript discusses insect- and fungal (oomycete)-derived small molecules. Either change the title so that it corresponds with the content, or reduce the content.
  • What was the motivation behind including sphingoid bases, caeliferins and bruchins? Why were other substances, such as bacterial volatiles, ignored? 
  • Lines 72–101 describe several examples of other receptors in plant cells. Considering the subsequent content, this seems unhelpful to me, as it adds further aspects to the manuscript that are neither interesting nor relevant to the subsequent discussion.
  • I found chapters 2, 3 and 4 difficult to read. While it is useful to introduce the different small molecules, a review should not describe the results of selected studies one by one. It would be better to provide an overview of the different, often non-congruent, findings in a table, summarising where studies overlap or contradict each other.
  • Furthermore, these three chapters conflate induced responses and perception. In particular, the first part ('Plant responses to GLVs and other signalling molecules') has been covered elsewhere (e.g. Engelberth, 2021; Matsui and Engelberth, 2022; Matsui and Koeduka, 2016; Ameyer et al., 2017), and the individual subchapters also reiterate the roles these molecules play in stressful conditions, particularly in chapters 3 and 4.
  • Chapters 5, 6 and 7 present some new findings based on the authors' predictions, which are interesting. However, there is some overlap in content with previous chapters here again. I recommend restructuring the chapters by merging similar sections on the molecules, their roles and (lack of) receptors.
  • Please reduce redundancy in the conclusions and shorten the text.
  • The authors highlight the importance of 'secondary organic aerosols'; based on the chemical structures of the molecules, it is likely that ozone, nitric oxide, and other pollutants alter their structure, as demonstrated for a variety of pheromones, for example. Additionally, consider that plants can modify GLVs emitted by neighbouring plants (e.g. Sugimoto et al., 2014) and insect elicitors, such as FACs (e.g. van Doorn et al., 2010).

Round 2

Reviewer 2 Report

Comments and Suggestions for Authors

Thank you for revising the manuscript and your detailed replies.

I have one additional comment. Please check the headings and subheadings in Chapter 2. a subheading is not necessary and rather irritating if there is only one subchapter.

The formatting of Table 3 needs to be corrected.

Author Response

Comment 1: Thank you for revising the manuscript and your detailed replies. I have one additional comment. Please check the headings and subheadings in Chapter 2. a subheading is not necessary and rather irritating if there is only one subchapter.

Response: Thanks for pointing that out. We really appreciate the work put into the reviews!

We changed Chapter headings by deleting ‘Fatty Acids’ and replacing it with the former subheading ‘Medium-chain Fatty Acids’. The former subheading was deleted (line 116).

Comment 2: The formatting of Table 3 needs to be corrected.

Response: We were not quite sure what the reviewer referred to. Perhaps the gap between the year of a reference and the number in brackets linking to the reference list? We had the number at the right side of the column. Maybe that looked awkward. Therefore, we moved the number directly after the year. We then revisited all other tables and changed them accordingly. We also noted that the column headings were not uniform in all tables. We changed them such that all first words in column headings are capitalized.

We generated a revised manuscript where the said changes are shown in red. The changes made for the previous revision are no longer shown in red.